# Modeling spatiotemporal dynamics of *Amblyomma americanum* questing activity in the central Great Plains

**Marlon E. Cobos**[1]*, **Taylor Winters**[2], **Ismari Martinez**[2], **Yuan Yao**[3], **Xiangming Xiao**[3], **Anuradha Ghosh**[4], **Kellee Sundstrom**[5], **Kathryn Duncan**[5], **Robert E. Brennan**[6], **Susan E. Little**[5], **A. Townsend Peterson**[1]

**1** Department of Ecology and Evolutionary Biology & Biodiversity Institute, University of Kansas, Lawrence, KS, United States of America, **2** Biodiversity Institute, University of Kansas, Lawrence, KS, United States of America, **3** School of Biological Sciences, Center for Earth Observation and Modeling, University of Oklahoma, Norman, OK, United States of America, **4** Department of Biology, Pittsburg State University, Pittsburg, KS, United States of America, **5** College of Veterinary Medicine, Oklahoma State University, Stillwater, OK, United States of America, **6** Department of Biology, Center for Interdisciplinary Biomedical Education and Research, University of Central Oklahoma, Edmond, OK, United States of America

* manubio13@gmail.com

**Data Availability Statement:** Tick data analyzed in this project and the code to reproduce all analyses are openly available at https://github.com/marlonecobos/Tick_KSOK. Other data used can be

## Abstract

Ticks represent important vectors of a number of bacterial and viral disease agents, owing to their hematophagous nature and their questing behavior (the process in which they seek new hosts). Questing activity is notably seasonal with spatiotemporal dynamics that needs to be understood in detail as part of mediating and mitigating tick-borne disease risk. Models of the geography of tick questing activity developed to date, however, have ignored the temporal dimensions of that behavior; more fundamentally, they have often not considered the sampling underlying available occurrence data. Here, we have addressed these shortfalls for *Amblyomma americanum*, the most commonly encountered tick in the central Great Plains, via (1) detailed, longitudinal sampling to characterize the spatiotemporal dimensions of tick questing activity; (2) randomization tests to establish in which environmental dimensions a species is manifesting selective use; and (3) modeling methods that include both presence data and absence data, taking fullest advantage of the information available in the data resource. The outcome was a detailed picture of geographic and temporal variation in suitability for the species through the two-year course of this study. Such models that take full advantage of available information will be crucial in understanding the risk of tick-borne disease into the future.

## Introduction

Tick-borne disease has come to be appreciated as a major public health concern in North America, owing both to its increasing frequency [1] and to the diversity of pathogens and types of illness that result [2–4]. Although the role of ticks in transmitting pathogens and causing disease in humans and animals has long been recognized [5], research attention has

accessed from the sources described in the Methods section.

**Funding:** This research is based upon work supported by the National Science Foundation under grant number OIA-1920946.

**Competing interests:** The authors have declared that no competing interests exist.

focused in largest part on select tick species (e.g., *Ixodes scapularis*) and pathogens (e.g., *Borrelia burgdorferi*, the causative agent of Lyme disease) [6–8]. However, there are various other tick species with medical significance. For example, *Amblyomma americanum*, the lone star tick, is capable of transmitting multiple zoonotic pathogens, including *Ehrlichia chaffeensis*, the primary causative agent of ehrlichiosis in humans, and is also associated with the development of alpha-gal syndrome, or red meat allergy, in humans [9]. This tick predominates in the eastern half of the United States (US) and has been moving westward over the last few decades [9]. This can, in part, be attributed to the continued biogeographical changes occurring across the US allowing its suitable habitat to expand and its activity time to lengthen [10]. In Oklahoma, *A. americanum* has displayed considerable changes in peak activity times in comparison to previous reports with active adults documented one month earlier than historical trends [11]. Some other studies have reported on outcomes of detailed sampling of tick populations, with summaries of phenology and documentation of range shifts [12, 13], but very few have studied regions with emerging or growing populations of *A. americanum*.

Models are used in disease ecology to generalize from sampling data and experimental data to establish broad patterns of distribution of key organisms [14], and to portray transmission risk to humans (or other species of interest) across geographic landscapes [15]. To date, these models have been in the form of mathematical, first-principles treatments of disease transmission systems [16], or of correlative models that seek associations between occurrence and aspects of environments and landscapes [17]. First-principles models have the advantage of comprehensive treatment of a transmission system and immunity to biasing effects of uneven sampling, but suffer from challenges in terms of estimating key parameter values and replicating fully the complexity of environments and geography [18]. Correlative models, on the other hand, are more universally applicable, as they require very simple and readily available data inputs, but suffer from myriad effects of biased and incomplete sampling [19].

One realm in which both sorts of models have failed to anticipate risk sufficiently, however, has been the spatial and temporal dynamics of environments and consequent suitability for disease-relevant organisms [20]. Although ideas of full temporal detail in correlative disease models have been explored preliminarily and discussed in the literature [20], they have not as-yet been implemented completely. Key tools with which to model such dynamic processes have been developed in the area of biodiversity science [21, 22], such that these steps are eminently feasible, awaiting only the combination of an appropriate dataset and sufficient computational time and resources.

In this contribution, we integrate three key elements—(1) a detailed dataset documenting tick distributions through time at 10 sites in Oklahoma and Kansas, (2) tools for explicit consideration of sampling in identifying and establishing key niche dimensions, and (3) fitting of correlative ecological niche models to the time-specific tick dataset taking into account both positive and negative occurrence data. We apply this novel combination of tools and inferences to the most common tick species across the Oklahoma and Kansas study region (i.e., *Amblyomma americanum*). Our main goal is to characterize climatic conditions that favor *A. americanum* questing activity, and generate weekly predictions across the central Great Plains. The outcome is a view of suitability through time for this species, across the entire region of study, which lays a foundation for summaries of the temporal and spatial dynamics of risk of transmission of key pathogens from ticks to humans or other species of interest. Our study serves as a proof of concept of these novel implementations of methods by which to process data into dynamic predictions of ecological phenomena. Public health officials and key stakeholders can use results generated using these techniques to make informed decisions toward the creation of disease-preventive measure.

## Materials and methods

Our workflow for this project can be summarized in five main steps. (1) We prepared records for the tick *A. americanum* to be used in ecological niche modeling using information obtained from detailed field sampling during 2020–2022. (2) We produced weekly environmental summaries in the form of raster layers to represent conditions in a time-specific manner, and associate such conditions with our tick records. (3) We used our records associated with environmental data to identify and characterize environmental conditions related to the occurrence of the tick (signals of niche). (4) We developed ecological niche models and created weekly predictions using the prepared data. Finally, (5) we performed post-modeling analyses to understand and represent ecological niche modeling results and uncertainty in our predictions. Scripts to reproduce all analyses are provided at https://github.com/marlonecobos/Tick_KSOK.

### Presence-absence data

Occurrence data were obtained for this set of analyses based on longitudinal sampling at 10 sites across Kansas and Oklahoma (Fig 1), as part of a large-scale project aimed at mapping risk of tick-borne diseases. Specific permits allowing us to collect ticks are not required in

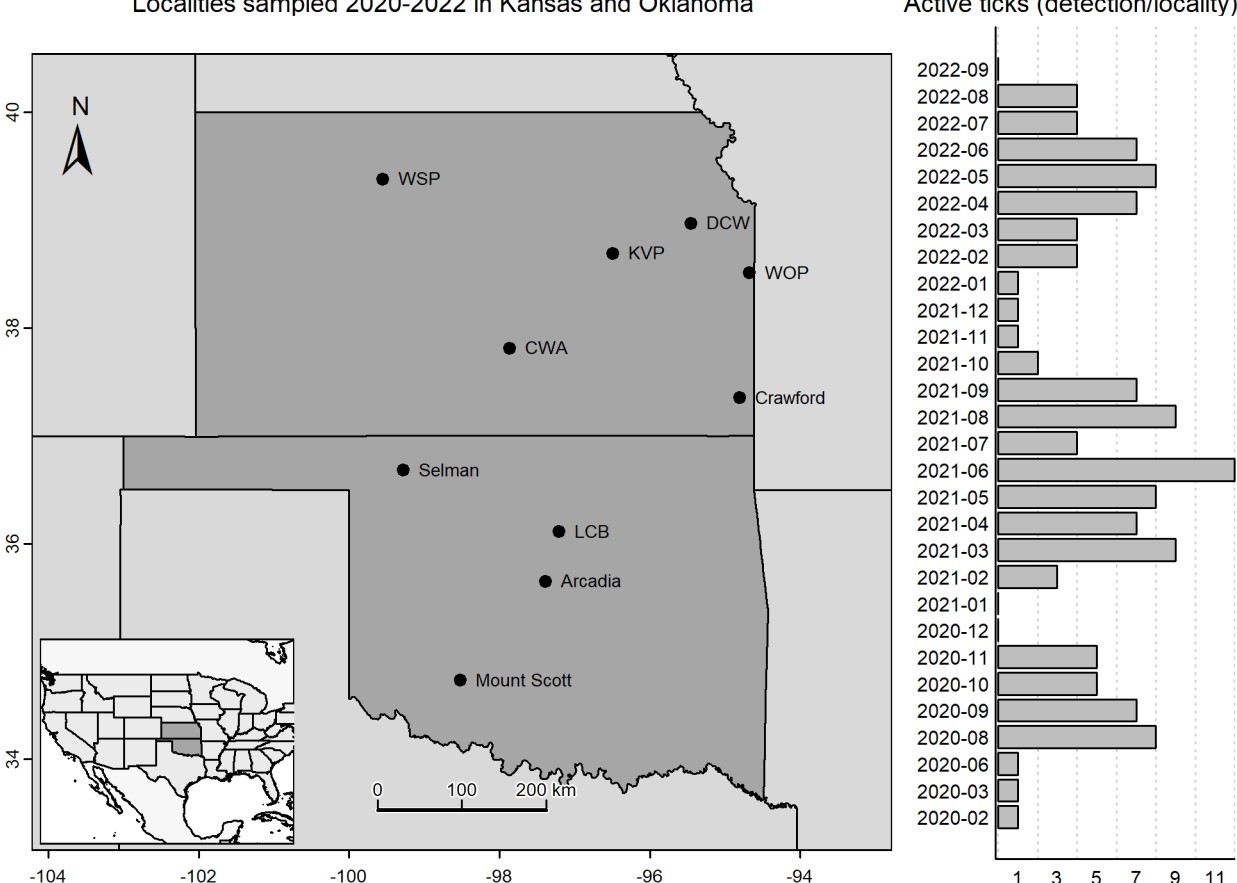

**Fig 1. Sampling sites at which field collections were used to produce time-specific models of questing activity for *Amblyomma americanum*.** Localities sampled 2020–2022 and monthly frequency of unique detections of active *A. americanum* per locality are presented. Inset shows the geographic position of the study area with respect to the Lower 48 United States.

these two states. Sampling was conducted between 1 August 2020 and 30 September 2022 (Fig 1), and consisted of intensive efforts to collect questing ticks in wooded and open areas at each site in the study, using a consistent sampling protocol combining flagging, dragging, and $CO_2$ traps. Sampling teams consisted of 3–7 persons, and visits lasted 3–4 hours, and sampling was conducted only under appropriate conditions for tick questing (temperature 2–32˚C, wind speed < 15 mi/h, and no precipitation or dew). In all, the sampling effort extended to 181 sampling visits across the 10 sites.

From the set of days x sites sampled, we obtained records of presences and absences for the tick species *A. americanum* (i.e., detection of questing ticks and non-detections). We assembled data sets in the form of all records of the species to understand general patterns of activity for this tick. As ticks at different life stages are found at varying rates in nature, and they serve as vectors for distinct pathogens, we also separated records by life stage (larva, nymph, adult). A presence record is a unique combination of species, day, month, year, and site; an absence is also a unique combination of the same fields, but for which the species was not detected. Presences and absences for each life stage of the tick were obtained similarly, but including life stage in the combination of fields. All data processing steps were performed using base functions in R [23].

## Environmental data

We obtained climatic data from the Daymet database (https://daac.ornl.gov/DAYMET/guides/Daymet_Daily_V4.html). We included the following variables: maximum air temperature (tmax), minimum air temperature (tmin), precipitation (prcp), shortwave radiation (srad), water vapor pressure (wvap), and day length (dayl). The variable snow water equivalent, also available in these datasets, was not included because no sampling was done under conditions with snow cover, and consequently this variable caused considerable confusion in results when included in preliminary, exploratory analyses.

The spatial resolution of the climatic data was about 1 km, and the summary raster layers were cropped to a rectangular extent covering the states of Kansas and Oklahoma (33.62–40.00˚ N, 103.00–44.43˚ W). We produced layers at a temporal resolution of 8 days, averaging over daily values, from 2020 to 2022 for an appropriate correspondence with the presence-absence data. The 8-day averaging periods resulted in such data 46 layers per year, although the last layer of each year covered only 5–6 days, depending on the year. We prepared 8-day averages of the Daymet data, as this is a common time interval at which satellite-derived data are analyzed, which could facilitate future exploration of other variables. We used the Julian day of the year for each weekly average layer as a means of naming layers uniquely. Variable processing analyses were done in Google Earth Engine [24].

We calculated Pearson's correlation coefficients among the 6 original variables, and detected high correlations between multiple pairs of variables (i.e., | $r$ | > 0.8; S1 Table). As a consequence, we used principal components analysis (PCA) to obtain sets of orthogonal predictors from the original variables. All six of the principal components deriving from the PCA were used in posterior analyses. This analysis was applied to data covering three full years (2020–2022), and we used the PCA to characterize every Julian week (8-day period) over the three years. We used the positive and negative occurrence records described above to extract values from raw environmental variables and the principal components according to the geographic coordinates and the Julian week of collection. This step created a dataset that characterized occurrences and non-occurrences in our tick-collection data in environmental dimensions specific in both time and space [21]. All of these raster processing procedures were performed with the R package terra [25].

## Niche signal exploration

A first analysis step was to test for a niche signal—in effect a bias in environmental space as regards which of our sampling events did *versus* did not detect *A. americanum*. If the species is present in all samples, or if detections are random with respect to environmental dimensions, then no niche signal is manifested across that time span and study region. Cobos and Peterson [22] presented multivariate (PERMANOVA) and randomization-based univariate tests for such niche signals. The PERMANOVA tests help to identify if the position and dispersion of the records of detection are similar to all of the records (detections and non-detections). Rejecting the PERMANOVA null hypothesis indicates that detections show a different environmental signal compared to all records (the sampling universe).

The univariate tests, on the other hand, assess niche signals via random resampling of the overall dataset to assess the random-distribution null hypothesis directly. Specifically, if one has an overall sample of size $N$, out of which a smaller number of samples (termed $x$) proved to be positive for a phenomenon (in this case, occurrence of questing *A. americanum*), then a large number (e.g., in this case 1000) of samples of size $x$ are drawn at random from the full set of $N$ sampling events. The mean, median, standard deviation, and range of these replicate samples in terms of environmental data values is then characterized, and the observed value among the real $x$ positive samples is compared to that distribution. Niche signal explorations were performed using the R package enmpa [26].

## Ecological niche model development

Ecological niche model calibration was done via methods that will contrast rather markedly with the usual approach to such questions (see, e.g., [27]), in view of the availability of high-quality presence and absence data from our tick sampling program across the two states (detection and non-detection of questing ticks). We used logistic generalized linear models (GLMs), which contrast populations of positive (detections) and negative (non-detections) occurrence data in predictions of probability of "presence" *versus* "absence." GLMs have been used previously in niche modeling applications [28–30], but usually based on contrasts between presence data and pseudoabsence data that were resampled from the parts of the study area that did not hold presences. We do not recommend or condone use of GLMs for presence-pseudoabsence data, but we do see them as appropriate and powerful approaches when high-quality presence-absence data are available.

Our GLM calibration approach consisted of testing a large suite of candidate models representing all possible combinations of predictors derived from linear and quadratic responses of 6 principal component variables, for a total of 4095 candidate models. All candidate models were created and evaluated for statistical significance and predictive ability using a 10 $k$-fold partitioning approach (i.e., the method to split training and testing data). Two metrics, area under the curve of the receiver operating characteristic (ROC AUC; [31]) and true skill statistic (TSS; [32]) were used to assess performance and predictive power of candidate models. We also created candidate models with the whole set of data and used the Akaike information criterion (AIC; [33]) to assess model goodness-of-fit penalized by model complexity.

Once all candidate models were fitted and evaluated, we retained the subset of models that passed four filtering steps. This model selection process was based loosely on previous approaches to presence-only niche models [34], and have been proposed as a way to obtain models with biologically realistic responses, that perform better than random expectations, that have good predictive power, and that have good balance between model fit and complexity [26]. That is, we retained models (1) that had all quadratic predictor coefficients negative (i.e., variables responses curves were convex/unimodal); from among those models, we retained (2)

models that had AUC $\geq$ 0.5, and then kept only (3) those with TSS $\geq$ 0.4; finally, (4) we used AIC to choose as final models only the ones with AIC scores within 2 units of the minimum value among those that had passed the first three filters ($\Delta$AIC $\leq$ 2).

Once we had selected final models, we transferred each of the set of best models separately to all weekly environmental summaries for 2020–2022. We created a weighted average of all best model predictions to obtain a single consensus prediction per week based on the AIC weights calculated for selected models. All modeling steps were performed on data at the species level (i.e., all occurrences of the species) and for each individual life stage separately (larva, nymph, adult) using the R package enmpa [26].

### Post-modeling analyses

We explored and visualized model predictions in various ways. Given the vagaries of weather and its variation in individual days and weeks, we smoothed weekly predictions using a moving window approach, in which each week was recast as the average value of the week before, the focal week, and the following week. We also produced monthly averages from weekly predictions by creating averages of the weekly rasters across the months in which they fall, including partial membership of weeks that overlap between sequential months.

To identify areas in Kansas and Oklahoma with environmental conditions outside the ranges of conditions represented in the data used for models, we used the mobility-oriented parity metric (MOP; Owens et al. 2013). MOP analyses were applied to all environmental summaries for 2020–2022. Monthly summaries of MOP results were obtained similarly to monthly averages of model predictions; higher values in these summaries indicate that such areas have been outside calibration ranges for more of the month. MOP analyses were done using the R package mop [35]. All other analyses in this section were done using the R package terra.

## Results

Processing our tick sampling data as detailed above, we obtained 130 presence and 48 absence records of tick questing activity that were used in the models described below. The first two principal components (PCs) explained 66% and 19% of the variance, respectively, in the overall analysis (S2 Table). The first PC had high loadings associated with tmin, tmax, dayl, and vp; the second PC had a high loading associated with prcp (S3 Table). The third PC had high loadings for prcp and srad, whereas the fourth PC had a high loading from dayl.

### Niche signal tests

A first step in the sequence of analyses in this study is that of assessing whether any significant (non-random) bias exists in the occurrences of *A. americanum*, as compared with the distribution of sampling events, in environmental spaces. The multivariate PERMANOVA test indicated that indeed such an environmental bias exists: i.e., that consistent and statistically significant environmental biases separate occurrences and from the complete set of records in our sampling regime ($F = 2.53$, $P < 0.05$).

Exploring these environmental biases in occurrence of *A. americanum* in our samples with respect to individual environmental variables indicated significant biases for all environmental variables (Table 1). Both mean and median indicated that occurrences of the tick were under conditions of longer day lengths, higher solar radiation, warmer maximum and minimum temperatures, and higher vapor pressure. For precipitation and snow water equivalent, the differences were significant only in tests based on medians, but not for tests based on means. For daylength, solar radiation, and maximum and minimum temperatures, observed distributions

**Table 1. Summary of univariate tests of the existence of a distinguishable environmental bias (i.e., an ecological niche) of the tick species *Amblyomma americanum* relative to all of the sampling events in this study.** "Higher" and "lower" indicate that the observed value of the occurrences of the tick with regard to the environmental dimension in question fell in the upper or lower 2.5% of the null distribution, respectively.

| Variable | Niche position | | Niche breadth | |
| --- | --- | --- | --- | --- |
| | Mean | Median | Standard deviation | Range |
| Day length | higher | higher | lower | lower |
| Precipitation | – | higher | – | higher |
| Solar radiation | higher | higher | lower | lower |
| Maximum temperature | higher | higher | lower | – |
| Minimum temperature | higher | higher | lower | – |
| Vapor pressure | higher | higher | – | – |

were narrower (standard deviation and/or range) than null distributions (Table 1). Similar patterns were found for explorations performed with data separated by life-stage (S4 Table).

## Ecological niche models

Out of the 4095 candidate models that were evaluated for all individuals of *Amblyomma americanum*, 482 passed the initial filter of having no quadratic functions that were upwardly concave (and therefore bimodal and inappropriate as ecological niche models). Of those models, 455 and 425 passed the next two sequential filters of AUC and TSS. Finally, 10 models were within 2 AIC units of the minimum among the models that passed the first three filters (Table 2). These 10 models all included the first principal component as a predictor, plus 1–3 other components and at least one quadratic term. AUC values were uniformly above 0.91, and TSS values were all above 0.77 (Table 2).

The two principal component variables that had the most dramatic effects on the 10 best models were the first and fourth principal components (Fig 2). 3. Averaging across the 10 best models, all response curves with respect to each of the principal component axes were unimodal. In all cases, the maximum value of the curve was at or near one of the limits of environmental availability across the sampling events (Fig 2).

The 10 best models for questing activity of *A. americanum* showed shifting patterns of activity through the year and across the study area, which were at least associated with trends in temperature and water availability (vapor pressure; Fig 3 and S1 Fig). Increases in temperature after cold periods seem to drive increases in probability of questing activity; whereas

**Table 2. Ten "best" models selected in the process of model calibration and evaluation, showing predictor variables; "^2" indicates quadratic terms in a model.** Values for AUC, sensitivity, specificity, and TSS are averages deriving from the 10 *k*-fold evaluation process. Threshold indicates the value used to test model sensitivity and specificity. AUC = area under the receiver operative characteristic curve; TSS = true skill statistic; AIC = Akaike information criterion; Δ AIC = delta AIC; AIC γ = AIC weight.

| ID | Predictors | Threshold | AUC | Sensitivity | Specificity | TSS | AIC | Δ AIC | AIC γ |
| --- | --- | --- | --- | --- | --- | --- | --- | --- | --- |
| 1 | PC1, PC2, PC3, PC4, PC1^2 | 0.619 | 0.919 | 0.854 | 0.94 | 0.794 | 117.1 | 1.51 | 0.088 |
| 2 | PC1, PC3, PC4, PC1^2 | 0.670 | 0.921 | 0.839 | 0.96 | 0.799 | 115.6 | 0.00 | 0.188 |
| 3 | PC1, PC3, PC4, PC1^2, PC3^2 | 0.676 | 0.921 | 0.846 | 0.96 | 0.806 | 117.6 | 1.99 | 0.070 |
| 4 | PC1, PC3, PC4, PC1^2, PC6^2 | 0.678 | 0.918 | 0.839 | 0.96 | 0.799 | 117.5 | 1.88 | 0.073 |
| 5 | PC1, PC3, PC4, PC5, PC1^2 | 0.648 | 0.919 | 0.854 | 0.94 | 0.794 | 117.4 | 1.80 | 0.076 |
| 6 | PC1, PC3, PC4, PC6, PC1^2 | 0.662 | 0.917 | 0.846 | 0.96 | 0.806 | 116.2 | 0.62 | 0.138 |
| 7 | PC1, PC4, PC1^2 | 0.686 | 0.931 | 0.846 | 0.96 | 0.806 | 117.1 | 1.49 | 0.089 |
| 8 | PC1, PC4, PC5, PC1^2 | 0.712 | 0.936 | 0.846 | 0.96 | 0.806 | 117.2 | 1.57 | 0.086 |
| 9 | PC1, PC4, PC5, PC6, PC1^2 | 0.613 | 0.925 | 0.854 | 0.92 | 0.774 | 116.9 | 1.28 | 0.099 |
| 10 | PC1, PC4, PC6, PC1^2 | 0.632 | 0.914 | 0.846 | 0.94 | 0.786 | 117.1 | 1.46 | 0.091 |

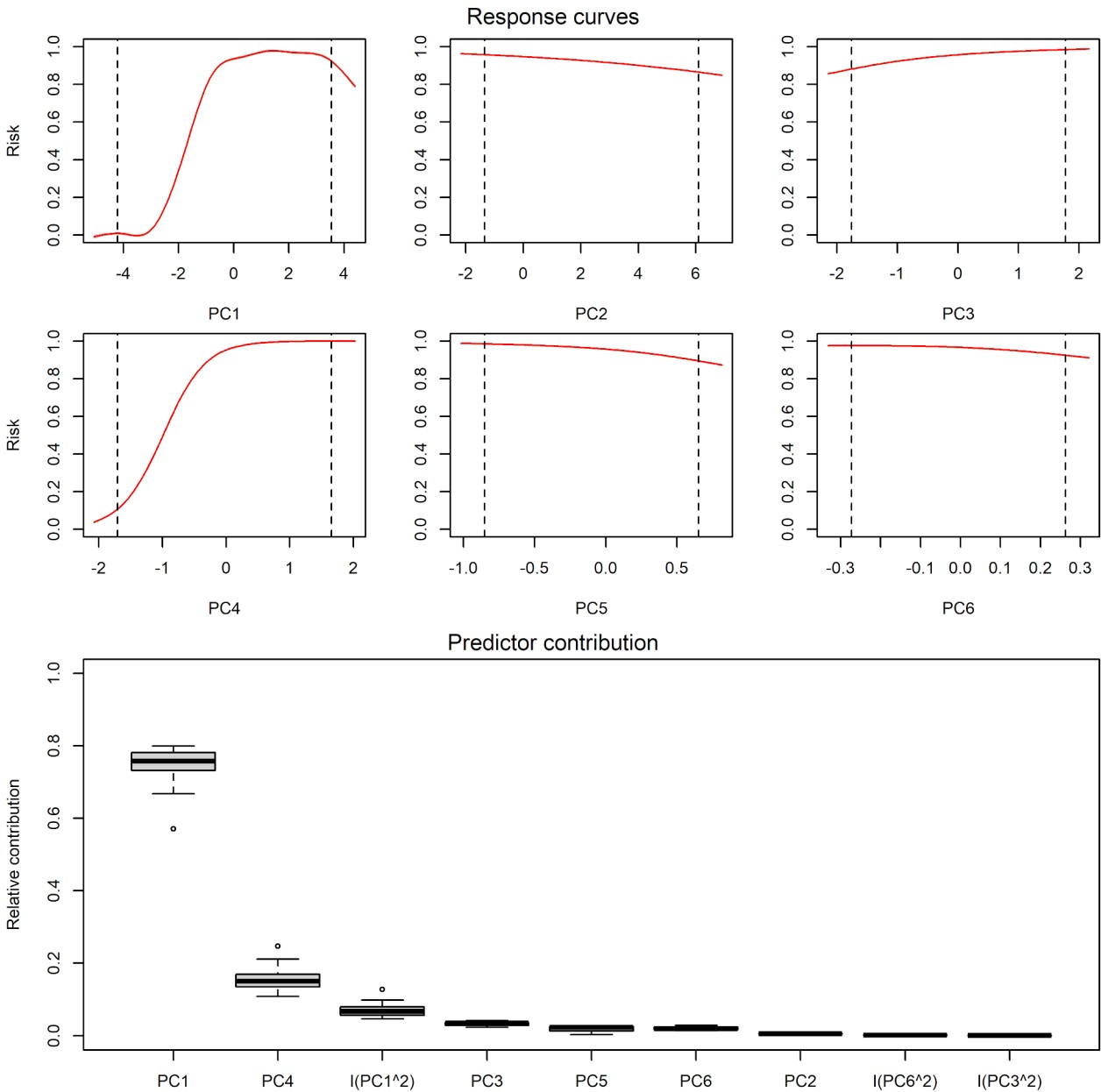

**Fig 2. Variable response curves and summary of all forms of predictor relative contribution used in models for *Amblyomma americanum* questing activity selected after model calibration.** PC = principal component. Response curves represent the probability of occurrence of questing activity given a value of the variable.

decreases in vapor pressure after humid periods appear to decrease such probabilities. Winter periods are ones of low or no activity for this species, but suitability builds through the course of the year in a southeast-to-north and southeast-to-west pattern. By about the beginning of April, suitability is high across the entire study region, and it remains high until the beginning of Fall (e.g., mid-September) when suitability declines somewhat over the entire region (Fig 4; weekly and monthly maps can be viewed as GIFs in S1, S2 and S4 Files). Finally, by about mid-October, suitability declines further to winter lows for the species. Applying the moving window approach to the weekly predictions did not change the general pattern of suitability and

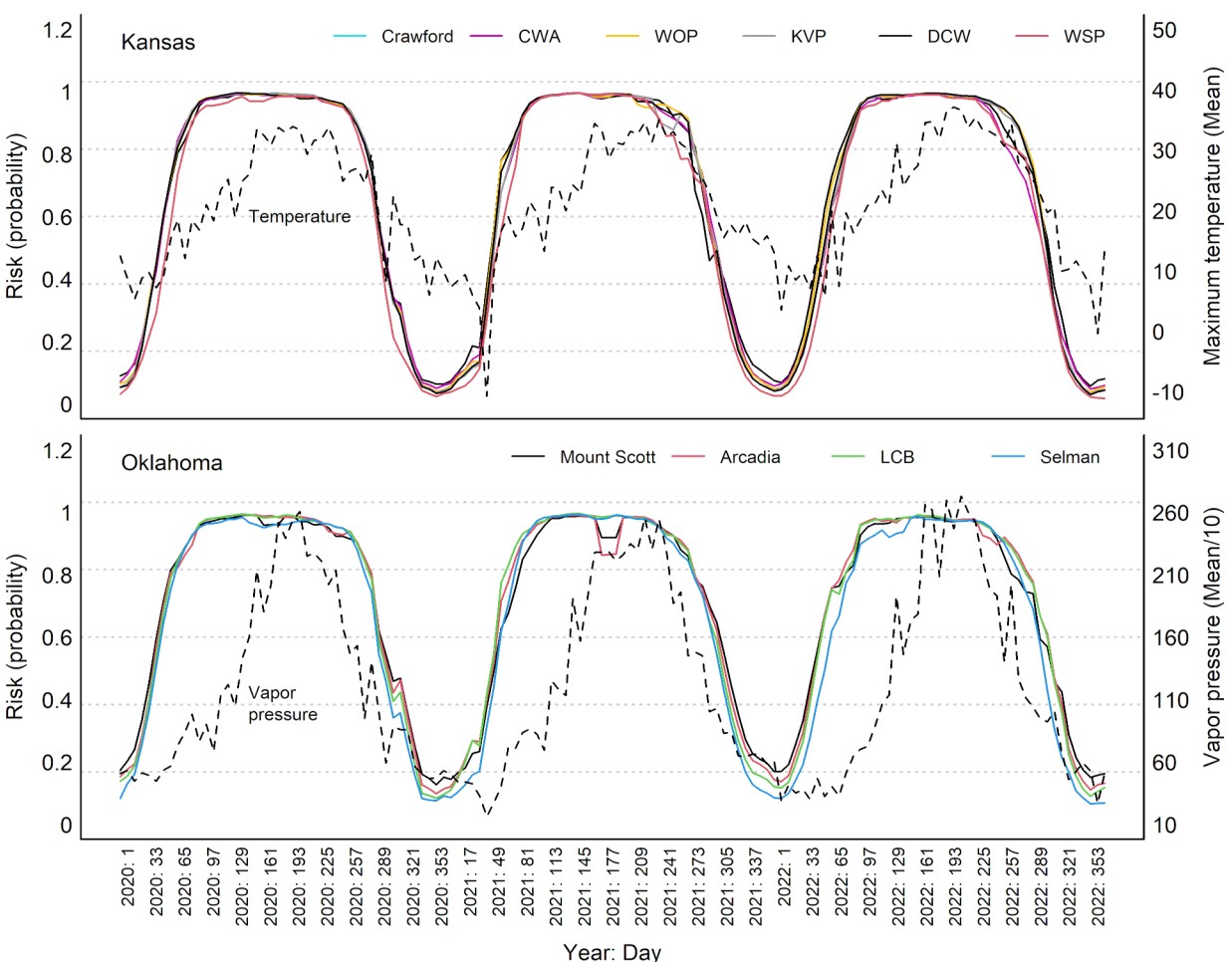

**Fig 3. Weekly dynamic of questing activity species-level predictions for *Amblyomma americanum* at the 10 localities sampled 2020–2022.** Predictions are shown for each of the sites sampled (solid lines); broken lines show 10-locality average patterns of variation in temperature (upper panel, Kansas) and vapor pressure (lower panel, Oklahoma).

its changes through the years or across the two states. Indeed, the only differences between the moving-window-smoothed results and the un-smoothed results were that the suitability trends were slightly more continuous and less jagged in the former compared with the latter (Fig 3, S1 Fig, and S1 and S2 Files).

The presence of non-analogous conditions (i.e., those outside of the range of conditions manifested at our sampling sites and dates) indicates that the interpretation of model predictions should be done carefully, as these are the result of model extrapolation. Extreme conditions were manifested principally in the extreme western and northwestern parts of the study area, as well as in the southeastern parts of the study area, depending on the time of year (Fig 5, S3 and S5 Files). Areas with non-analogous conditions were generally associated with combinations of dry and hot short-term climatic conditions.

Models for individual life stages of *A. americanum* had similar characteristics as the overall models (S5 Table). That is, selected models for all life stages passed all of the filters that we imposed on the candidate model set. As regards to predictor contribution to models, the first PC had high contribution in models selected for the distinct life stages; the fourth PC was an important predictor in models for Nymphs, whereas for Adult and Larva models, the third PC

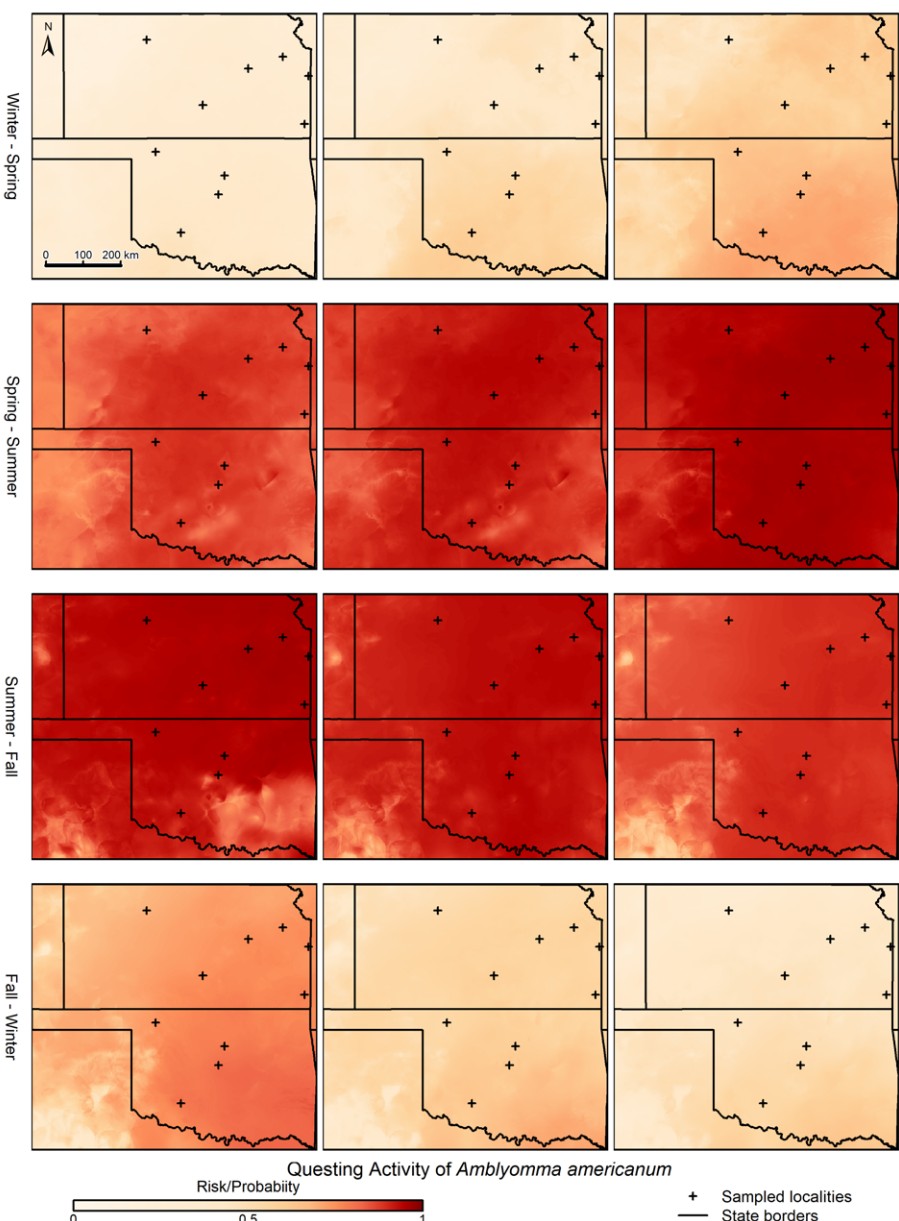

**Fig 4. Example geographic species-level predictions of weekly questing activity for *Amblyomma americanum* across the states of Kansas and Oklahoma, in 2020.** Only key weeks in which suitability changes as a consequence of seasonal environmental changes are shown. Black crosses show sampling sites (see Fig 1).

was a more relevant predictor (S2–S4 Figs). Predictor response curves varied across the distinct life stages but they were all unimodal.

In geographic terms, the models for larvae did not reveal strong or clear geographic patterns: if anything, the study region became more suitable (mildly, though) for *A. americanum* larvae later in the Summer and into the Fall (S6 and S7 Files). Patterns in models for nymphs were more clear, and mirrored those in the overall models: suitability rose markedly across the region in the Spring, waned somewhat in the Summer and Fall, and declined to near-nil in the Winter (S8 and S9 Files). Finally models for adult *A. americanum* only showed parallel

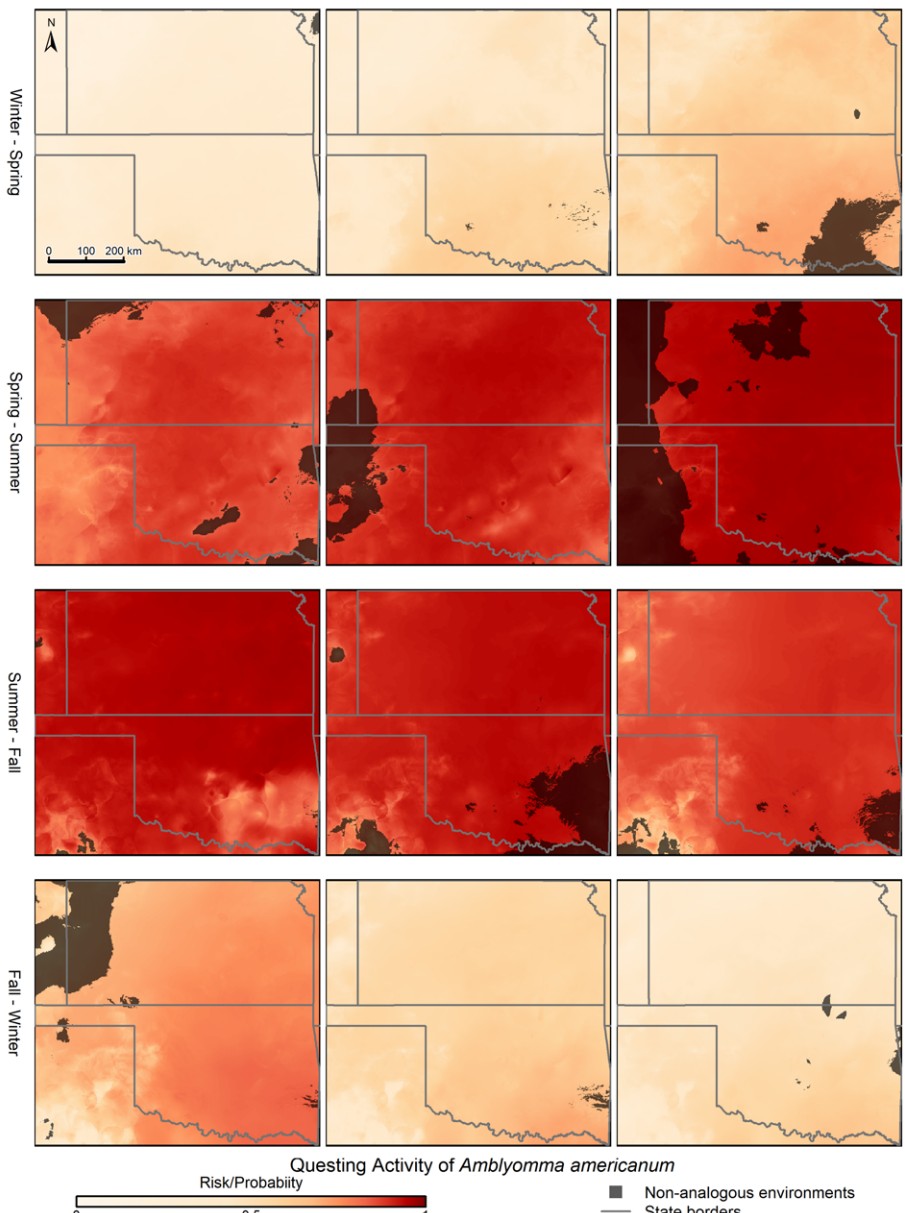

**Fig 5. Geographic representation of uncertainty in species-level model predictions across the study area deriving from model transfer to non-analogous environmental conditions.** Areas of uncertainty are visualized on examples of weekly questing activity predictions for *Amblyomma americanum* in the states of Kansas and Oklahoma, year 2020. Only key weeks in which suitability changes as a consequence of seasonal environmental changes are shown.

temporal trends to those of the nymphs, but showed a clear northwest-to-southeast pattern of rise in suitability in the Spring, as well as the reverse pattern in the decline in suitability in the Fall (S10 and S11 Files).

## Discussion

This contribution provides an initial, although detailed, exploration of the spatiotemporal dynamics of questing activity by *A. americanum* ticks in the central Great Plains. Our work

differs from other applications in that we are not characterizing environmental conditions suitable for the presence or persistence of *A. americanum* populations (e.g., [36]). Rather, we characterize short-term climatic conditions that are associated with tick questing activity, which may be more closely related to human exposure to tick-borne pathogens (e.g., *Ehrlichia*, *Rickettsia*, *Heartland virus*, etc.). The maps presented here should not be considered directly as maps of risk of tick-borne disease, but they represent a step forward in translating results from these types of models into representations of risks from exposure to disease vectors.

That *A. americanum* is currently distributed broadly across the study region (i.e., the states of Kansas and Oklahoma) is very well known: see, e.g., the ubiquitous distribution of the species in all of Oklahoma except the far-western Panhandle [37]. However, the seasonal activity of this tick species is not constant, and evidence shows it may be evolving as continued biogeographical changes occur across this region [11]. Using data collected over a period of two years under a consistent and repeated protocol, however, we were able to identify environmental combinations that are associated with high probability of questing activity by this species. Our time-specific predictions thus offer novel insights into the details of how tick questing activity varies across space and through the seasons.

Environmental conditions in our area of study vary dramatically, and play a crucial role in driving suitability estimates emerging from our models. Temperatures in the area range from -24.3˚C to 42.8˚C, and precipitation ranges from nil to as much as 36.9 mm in an 8-day period. Our models were able to identify environmental combinations that had consistent associations with increased *versus* decreased tick questing activity. As ticks rely on external temperature to regulate their metabolic rates, this variable is crucial for their development; however, extreme high temperature values can also promote faster dessication, reducing survival rates [38]. Our models show that, after peaks of temperature are reached in the area, A. americanum activity starts to decrease (Fig 3). Humidity is also a key environmental factor, since ticks lose moisture rapidly in dry conditions [39, 40]. Our model predictions also show that vapor pressure (a proxy of humidity) is highly related to probability of tick activity, with diminishing questing probabilities when this variable decreases in late summer (Fig 3). These relationships also revealed interesting patterns of spatial variation in suitability through time. For instance, the seasonal geographic patterns for questing activity levels obtained for different life stages of *A. americanum* could be associated with the expected transition of life cycle stages and the movement of established populations westward.

To our knowledge, this is the first time that levels of risk associated with *A. americanum* questing activity are analyzed and mapped via a time-specific, correlative workflow (but see Daniel et al. [41], for a similar example in Europe). Clearly, mathematical transmission models have seen considerable exploration regarding mosquito-borne diseases [42], and are now seeing increasing analysis for tick-borne diseases as well [43–45]. However, such first-principles, mechanistic models often suffer from the need for precise estimates of many parameter values [18]; in addition, such models have not been developed at large scales, such that population processes in one cell under one set of environmental conditions are linked to processes in adjacent or nearby cells under other environmental conditions (but see Cecilia et al. [46]). Correlative models such as those explored herein have the advantage of near-universal applicability, requiring only occurrence and nonoccurrence information, although they are clearly vulnerable to effects of bias in sampling and in representation of environments across regions [21]. In this sense, the correlative, data-driven models that we explore herein may offer a useful, complementary view of tick-borne disease risk that can allow predictions under conditions that would be difficult for application of mechanistic models.

Much work remains to be done, before models of this sort (or from other modeling avenues) can be used with confidence as risk maps for disease. For example, some of the more

interesting predictions resulting from our models should be tested via independent sampling and detailed statistical testing of model predictions. In addition, tick activity depends on changes in climatic conditions at fine temporal and spatial resolution (i.e., sudden changes in wind speed or day-to-day fluctuations of temperature in specific areas can affect profoundly the chances to observe questing behavior [38–40]). Using methods similar to the ones presented here, generating whether-like forecasts of tick activity is, therefore, potentially achievable. However, we anticipate that successful implementations in such directions would require further investigation in two main avenues: (1) identifying procedures by which to gather and process data (detection/non-detection of active ticks) that can be properly associated with weather information; and (2) defining modeling methods to characterize conditions that favor questing activity accounting for medium-term conditions that favor population growth. More generally, attention should be paid to factors molding causal pathways between suitability for tick populations or tick questing activity and actual patterns of disease occurrence in humans and animals. However, the information presented herein (more specifically our models) can be a crucial first step toward further explorations to understand how or if recent environmental changes can explain tick-reports in non-endemic areas—as the risk of tick-borne infections is growing. Ultimately, predictions deriving from similar models that can consider detection/non-detection and medium- to short-term climatic conditions, in concert, can help medical and veterinary professionals as well as government agencies to design better-informed public health actions.

## Supporting information

**S1 Fig. Non-smoothed weekly dynamic of questing activity species-level predictions for *Amblyomma americanum* at the 10 localities sampled 2020–2022.** Predictions are shown for each of the sites sampled (solid lines); broken lines show 10-locality average patterns of variation in temperature (upper panel, Kansas) and vapor pressure (lower panel, Oklahoma).
(EPS)

**S2 Fig. Variable response curves and summary of all forms of predictor relative contribution used in models for larvae *Amblyomma americanum* questing activity selected after model calibration.** PC = principal component. Response curves represent the probability of occurrence of questing activity given a value of the variable.
(EPS)

**S3 Fig. Variable response curves and summary of all forms of predictor relative contribution used in models for nymph *Amblyomma americanum* questing activity selected after model calibration.** PC = principal component. Response curves represent the probability of occurrence of questing activity given a value of the variable.
(EPS)

**S4 Fig. Variable response curves and summary of all forms of predictor relative contribution used in models for adult *Amblyomma americanum* questing activity selected after model calibration.** PC = principal component. Response curves represent the probability of occurrence of questing activity given a value of the variable.
(EPS)

**S1 Table. Summary of Pearson product-moment correlations among environmental variables.** Names of variables are as follows: maximum air temperature (tmax), minimum air temperature (tmin), precipitation (prcp), shortwave radiation (srad), water vapor pressure (wvap),

and day length (dayl).
(DOCX)

**S2 Table. Summary of variance summarized by each of the principal components (PCs).**
(DOCX)

**S3 Table. Summary of loadings of raw climatic variables on each of the 6 principal components (PCs).** Names of variables are as follows: maximum air temperature (tmax), minimum air temperature (tmin), precipitation (prcp), shortwave radiation (srad), water vapor pressure (wvap), and day length (dayl).
(DOCX)

**S4 Table. Summary of univariate tests for the existence of a distinguishable environmental bias (ecological niche) of the tick species *Amblyomma americanum*, for each life stage of the tick species separately, relative to all of the sampling events in this study.** "Higher" and "lower" indicate that the observed value of the occurrences of the tick with regard to the environmental dimension in question fell in the upper or lower 2.5% of the null distribution, respectively.
(DOCX)

**S5 Table. "Best" models selected in the process of model calibration and evaluation performed per life stage.** Predictor variables are shown; "^2" indicates quadratic terms in a model. Values for AUC, sensitivity, specificity, and TSS are averages deriving from the 10-kfold evluation process. Threshold indicates the value used to test model sensitivity and specificity. AUC = area under the receiver operative characteristic curve; TSS = true skill statistic; AIC = Akaike information criterion; Δ AIC = delta AIC; AIC γ = AIC weight.
(DOCX)

**S1 File. GIF animation of model-based predictions of suitability for questing activity by *Amblyomma americanum* ticks across Kansas and Oklahoma in 2020–2022.** Time steps are 8-day periods throughout each year.
(GIF)

**S2 File. GIF animation of non-smoothed model-based predictions of suitability for questing activity by Amblyomma americanum ticks across Kansas and Oklahoma in 2020–2022.** Time steps are 8-day periods throughout each year.
(GIF)

**S3 File. GIF animation of extrapolative conditions in models attempting to summarize suitability for questing activity by Amblyomma americanum ticks across Kansas and Oklahoma in 2020–2022.** Time steps are 8-day periods throughout each year.
(GIF)

**S4 File. GIF animation of model-based predictions of suitability for questing activity by *Amblyomma americanum* ticks across Kansas and Oklahoma in 2020–2022.** Time steps are months throughout each year.
(GIF)

**S5 File. GIF animation of model-based predictions of suitability for questing activity by *Amblyomma americanum* ticks (ramp of red) with extrapolative conditions (ramp of gray) overlaid across Kansas and Oklahoma in 2020–2022.** Time steps are months throughout each year.
(GIF)

**S6 File. GIF animation of model-based predictions of suitability for questing activity by *Amblyomma americanum* larvae across Kansas and Oklahoma in 2020–2022.** Time steps are 8-day periods throughout each year.
(GIF)

**S7 File. GIF animation of model-based predictions of suitability for questing activity by *Amblyomma americanum* larvae across Kansas and Oklahoma in 2020–2022.** Time steps are months throughout each year.
(GIF)

**S8 File. GIF animation of model-based predictions of suitability for questing activity *by Amblyomma americanum* nymphs across Kansas and Oklahoma in 2020–2022.** Time steps are 8-day periods throughout each year.
(GIF)

**S9 File. GIF animation of model-based predictions of suitability for questing activity by *Amblyomma americanum* nymphs across Kansas and Oklahoma in 2020–2022.** Time steps are months throughout each year.
(GIF)

**S10 File. GIF animation of model-based predictions of suitability for questing activity *by Amblyomma americanum* adults across Kansas and Oklahoma in 2020–2022.** Time steps are 8-day periods throughout each year.
(GIF)

**S11 File. GIF animation of model-based predictions of suitability for questing activity by *Amblyomma americanum* adult ticks across Kansas and Oklahoma in 2020–2022.** Time steps are months throughout each year.
(GIF)

## Acknowledgments

We thank a large suite of individuals who participated in the field sampling trips that resulted in the data that were key inputs into our models. A. Alkishe, D. Romero-Álvarez, C. Nuñez-Penichet, B. J. Wiens, P. Campbell, A. Fulk, S. Tusuubira, and A. Adeboje, from the University of Kansas; B. Letterman, S. Nippoldt, A. Dasgupta, C. Utley, H. Belgum, and L. Carrico, from Pittsburg State University; and M. Lineberry, K. McClung, S. Struble, and A. Grant, from Oklahoma State University.

## Author Contributions

**Conceptualization:** Marlon E. Cobos, A. Townsend Peterson.

**Data curation:** Marlon E. Cobos, Taylor Winters, Ismari Martinez, Yuan Yao, Xiangming Xiao, Anuradha Ghosh, Kellee Sundstrom, Kathryn Duncan, Robert E. Brennan, Susan E. Little, A. Townsend Peterson.

**Formal analysis:** Marlon E. Cobos, Taylor Winters, Ismari Martinez, Yuan Yao.

**Funding acquisition:** Susan E. Little, A. Townsend Peterson.

**Investigation:** Marlon E. Cobos, A. Townsend Peterson.

**Methodology:** Marlon E. Cobos, A. Townsend Peterson.

**Writing – original draft:** Marlon E. Cobos, A. Townsend Peterson.

**Writing – review & editing:** Marlon E. Cobos, Xiangming Xiao, Anuradha Ghosh, Kellee Sundstrom, Kathryn Duncan, Robert E. Brennan, Susan E. Little, A. Townsend Peterson.

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
