## [Decision Letter · Decision Letter 0]

9 Jun 2024

PONE-D-24-14393Spatiotemporal dynamic models of Amblyomma americanum questing activity in the Central Great PlainsPLOS ONE

Dear Dr. Cobos,

Thank you for submitting your manuscript to PLOS ONE. After careful consideration, we feel that it has merit but does not fully meet PLOS ONE’s publication criteria as it currently stands. Therefore, we invite you to submit a revised version of the manuscript that addresses the points raised during the review process.

The reviewers agree that your study is of interest, but both have concerns about the modeling approach that you used. Additionally, it is not clear how your work addresses and fits into the bigger picture of climate change and expansion of tick vectors of pathogens. 

We look forward to receiving your revised manuscript.

Kind regards,

Ulrike Gertrud Munderloh, Ph.D.

Academic Editor

PLOS ONE

Journal Requirements:

2. In your Methods section, please provide additional information regarding the permits you obtained for the work. Please ensure you have included the full name of the authority that approved the field site access and, if no permits were required, a brief statement explaining why

3. Thank you for stating the following financial disclosure: "This research is based upon work supported by the National Science Foundation under grant number OIA-1920946." 

Reviewers' comments:

Reviewer's Responses to Questions

**Comments to the Author**

1. Is the manuscript technically sound, and do the data support the conclusions?

Reviewer #1: Partly

Reviewer #2: Partly

2. Has the statistical analysis been performed appropriately and rigorously? 

Reviewer #1: Yes

Reviewer #2: Yes

3. Have the authors made all data underlying the findings in their manuscript fully available?

Reviewer #1: Yes

Reviewer #2: Yes

4. Is the manuscript presented in an intelligible fashion and written in standard English?

Reviewer #1: Yes

Reviewer #2: Yes

5. Review Comments to the Author

Reviewer #1: Intro: OK

Methods:

Figure 1:

Add a legend, a North Arrow and a scale to the map. A background depicing major geographical elements in the study region of landscape cover would be interesting. The x-axis of the histogram at the top is on an inversed chronological order. It would be good to have it started in year 2020.

L139-140 Justify the choice of the 8 days averaging for Daymet data. This is unusual: we are more used to have weekly average

L 136-137. Remove the mention of Daymet data before 202 if you don’t use them If you do, please clarify the way you used them.

L148 please specify the results of the PCA used into the absence/presence data analysis (maybe first and second components only?).

L160 what is the difference between a PERMANOVA test and a cluster analysis?

L 199-203 would it be possible to clarify the reason why this model selection approach is valid also for your presence AND absence data?. It would be also good to give some details about the rational behind each steps of this model selection process.

L212-225 I think this section is a bit week. It misses validation (comparison with independent dataset) and/or a sort of sensitivity analysis.

Results

OK

Discussion

The discussion clearly miss global and in details interpretation of the models predictor in terms of tick biology.

L335 the models used Daymet data that are weather variable. When saying ‘environmental’ variables we would expect to have landuse/landcover predictors in the model, but it is not the case. I suggest to clarify/modify the sentence here.

L343 there seems to be a ‘is’ that is not necessary in the sentence

L350-351 The usage of this model for the tracking of the ticks expansion would have required an assessment of the model sensitivity in space. Considering the limited number of ticks sampling sites (n=10) used in this study and the absence of validation/comparison with external dataset, I would recommend removing this suggestion.

One on the possible utility of the model developed is its usage as forecasting tool (using weather prediction). I suggest author to integrate this if possible.

In the legend of Figure 4 , 5 and 6, specify the North Arrow, the stage of the tick concerned by the prediction, and a spatial scale

Supplementary files: I wasn’t able to access them

Reviewer #2: This is an interesting article, which uses a statistical modelling approach to identify environmental suitability for questing activity of the tick Amblyomma americanum in a focal area of the US. As this location is well within broad suitability, the models are not identifying suitability for tick population persistence. Activity of this tick is determined by temperature and humidity but also possibly effects of daylength switching on and off diapause, so the approach is intrinsically interesting. I have some concerns about the manuscript in its present form.

Major points:

The selected temperature, humidity, daylength explanatory variables are all going to be highly correlated, but I don’t see accounting for this. Daylength as a continuous variable would only have an effect on activity as a proxy of temperature and possibly humidity. I think effects of daylength on switching on and switching off activity (behavioural diapause) would need to be modelled using some kind of threshold.

The authors use smoothing to ‘iron out’ vagaries of weather – but weather at the time of sampling is likely very important for tick activity – i.e. if the mean temperature is 20C over a week, that would be temperature suitable for tick activity, but if you happen to visit the site on one day when the temperature is 8C you might not find questing ticks – and the 8C is the determinant of that result – not the average 20C.

I am not convinced of the value of the ‘Niche signal test’ analysis – we know a priori that there are temperature and humidity limits on tick activity and these occur at specific times of the year that correlate with daylength.

More minor points:

The article would be improved by a more careful explanation of the objective – i.e. identifying when and where tick activity is occurring. But who would use this information and what for?

Also some alteration of the introduction is needed to explain the focus on A. americanum. PlosOne is an international journal and there are many tick species of public health importance beyond I. scapularis.

The importance of different tick instars as vector of different pathogens should be identified – nymphs and adults (but particularly nymphs which are more numerous than adults) for pathogens with no transovarial transmission, but larvae also for pathogens with transovarial transmission.

Are the numbers of ticks collected at a particular site visit independent of those at the previous visit? If not this needs justification, and if so it needs some kind of accounting in the GLMs.

I think the comparison of observed tick questing activity and model predictions should be in the main paper.

I’m not sure of the value of the mapped ‘extreme conditions’ in Fig 5 as these are only present at some times of the year and don’t describe conditions that result in absence of the ticks.

Figure 1 – the x-axis of time for the bar chart is going right to left rather than the expected left to right.

6. PLOS authors have the option to publish the peer review history of their article (what does this mean?). If published, this will include your full peer review and any attached files.

Reviewer #1: No

Reviewer #2: No

---

## [Author Response · Author response to Decision Letter 0]

2 Oct 2024

Dear Prof. Munderloh,

Thank you for your quick and useful editorial consideration of the manuscript, “Modeling spatiotemporal dynamics of Amblyomma americanum questing activity in the Central Great Plains.” We appreciate the useful comments and suggestions provided by the reviewers. We have made appropriate clarifications and modifications in our manuscript in response to those comments and suggestions. We believe that our manuscript has improved substantially, and we hope that you will now find it suitable for publication in PLoS ONE.

Please, find below detailed responses to the comments of the reviewers.

Sincerely,

The authors

Editor:

The reviewers agree that your study is of interest, but both have concerns about the modeling approach that you used. Additionally, it is not clear how your work addresses and fits into the bigger picture of climate change and expansion of tick vectors of pathogens. 

Response: We understand the concerns. We would like to note that this analysis is not on climate change. Rather, the project set out to understand spatiotemporal patterns of transmission risk of tick-borne pathogens to humans across the central Great Plains. We have modified our text in the introduction to make that clearer. Lines 75–77 and 80–81.

Response: We have made sure our manuscript meets PLOS ONE’s style requirements.

Response: Permits to collect ticks are not required in Kansas and Oklahoma. We have provided a brief statement to that effect in our Methods section.

3. Thank you for stating the following financial disclosure: "This research is based upon work supported by the National Science Foundation under grant number OIA-1920946." 

Response: The statement is correct. We have included this amendment in the cover letter. 

Reviewer 1:

Intro: OK

Methods:

Figure 1: Add a legend, a North Arrow and a scale to the map. A background depicting major geographical elements in the study region of landscape cover would be interesting. The x-axis of the histogram at the top is on an inversed chronological order. It would be good to have it started in year 2020.

Response: We have included the north arrow and scale bar as suggested. In the PDF formatted by the journal system, the figure is rotated 90° counterclockwise, which is why the date order appears to be in non-chronological order. Once the figure is rotated, dates are in chronological order from bottom to top. 

L139-140 Justify the choice of the 8 days averaging for Daymet data. This is unusual: we are more used to have weekly average

Response: We prepared 8-day averages as this is a common interval of time at which satellite-derived data are made available. Use of this time interval could facilitate the exploration of other variables in the future. This statement has been included in the text in lines 135–136.

L 136-137. Remove the mention of Daymet data before 2020 if you don’t use them If you do, please clarify the way you used them.

Response: We now only mention Daymet data 2020–2022. Lines 132–133.

L148 please specify the results of the PCA used into the absence/presence data analysis (maybe first and second components only?).

Response: We have added a short description of the results from the PCA in our text, as part of results. Lines 221–226.

L160 what is the difference between a PERMANOVA test and a cluster analysis?

Response: The main difference between these two statistical methods is that PERMANOVA is a hypothesis-testing method focused on comparing predefined groups, whereas cluster analysis is an exploratory technique used to discover natural groupings within a dataset. Regarding the methods used: PERMANOVA uses permutation tests to assess group differences based on a distance matrix. Cluster analysis uses various algorithms to find and from groups based on similarity or distance. If we consider the outputs, the PERMANOVA provides a test statistic and p-value; cluster analysis provides information that is used to visually represent groups.

L 199-203 would it be possible to clarify the reason why this model selection approach is valid also for your presence AND absence data?. It would be also good to give some details about the rational behind each steps of this model selection process.

Response: We have added to our text an explanation of why this selection approach is valid for presence-absence data, and included a reference for this statement. Lines 191–193.

L212-225 I think this section is a bit week. It misses validation (comparison with independent dataset) and/or a sort of sensitivity analysis.

Response: The section the reviewer refers to (Post-modeling analyses) describes: (1) simple averaging steps that help visualize the behavior of predictions after considering the effects of the temporally close predictions; and (2) an application of a well-established method in the field to detect regions where environments non-analogous to model-training conditions exist. The analyses performed do not require validation or sensitivity analyses.

Discussion:

The discussion clearly miss global and in details interpretation of the models predictor in terms of tick biology.

Response: We have added a couple of sentences about the relevance of the predictors used related to tick biology and ecology. Lines (357-364)

L335 the models used Daymet data that are weather variables. When saying ‘environmental’ variables we would expect to have landuse/landcover predictors in the model, but it is not the case. I suggest to clarify/modify the sentence here.

Response: We agree. We have modified our text, and now the sentence refers to short-term climatic conditions. Lines (337-338)

L343 there seems to be a ‘is’ that is not necessary in the sentence

Response: The word “is” is correctly placed in the sentence.

L350-351 The usage of this model for the tracking of the ticks expansion would have required an assessment of the model sensitivity in space. Considering the limited number of ticks sampling sites (n=10) used in this study and the absence of validation/comparison with external dataset, I would recommend removing this suggestion.

Response: We have removed that part of the sentence. Lines 350–351

One of the possible utility of the model developed is its usage as forecasting tool (using weather prediction). I suggest author to integrate this if possible.

Response: We have integrated ideas on the potential of further developing these tools to use them as a forecasting tool that uses weather conditions. In this text, we discuss important factors to consider and some limitations to be expected. Lines 385–393.

In the legend of Figure 4 , 5 and 6, specify the North Arrow, the stage of the tick concerned by the prediction, and a spatial scale

Response: The reviewer asks for changes in figures 4, 5, and 6. As we do not have Figure 6, we assume this relates to Figures 3–5. We have added north arrows and scale bars to the pertinent figures. All these figures represent prediction results at the species level (i.e., all stages combined). We have clarified that in the text describing the figures.

Supplementary files: I wasn’t able to access them

Response: We are sorry and concerned to hear that. We were successful in downloading the files when clicking on the access/download links in the manuscript's PDF file. Another option to download the files is to right-click and copy the links and then paste them into your internet browser. Hopefully, this time it works.

Reviewer 2:

The selected temperature, humidity, daylength explanatory variables are all going to be highly correlated, but I don’t see accounting for this. 

Response: We performed a principal component analysis (PCA) to deal with correlated predictors in our modeling steps. PCA generates a new set of orthogonal axes (non-correlated) which were used as predictors in our modeling steps. See lines 140-143.

Daylength as a continuous variable would only have an effect on activity as a proxy of temperature and possibly humidity. I think effects of daylength on switching on and switching off activity (behavioural diapause) would need to be modelled using some kind of threshold.

Response: Thanks for the suggestion. We agree that the reason that daylength is informative in our explorations can be related to how this variable covaries with others. However, we are not certain of how the effect of daylength should be modeled—that is to say, we feel that we do not have enough information a priori to be able to choose a threshold wisely. Considering our sample size and the temporal resolution of our data, then, we believe it is safer to use the variables as they are (i.e., without thresholding), as a threshold response can generate more complex, and perhaps better-fitted models, but not necessarily more biologically realistic ones. Consequently, we chose to keep our models the way they are, as using daylength with a threshold response would requiring more detailed explorations of the effects of this variable on tick activity, and even experimental manipulations to avoid biased characterizations.

The authors use smoothing to ‘iron out’ vagaries of weather – but weather at the time of sampling is likely very important for tick activity – i.e. if the mean temperature is 20C over a week, that would be temperature suitable for tick activity, but if you happen to visit the site on one day when the temperature is 8C you might not find questing ticks – and the 8C is the determinant of that result – not the average 20C.

Response: We completely agree with the reviewer, tick activity is very much driven by weather at fine temporal resolution. Our main argument for developing models at the temporal resolution of weeks was that this more or less matched the temporal resolution of climatic conditions with which we felt confident to associate our data (active tick records), considering that the spatial resolution of layers was ~1 km. Our goal, hence, was not to present predictions at a finer temporal resolution, but rather, coarser averages of what the risk of finding active ticks would look like in the area. Our smoothing method allows us to be more cautious about predictions which we do not consider a limitation. However, we have added more text to the discussion about this topic and how future research efforts can lead to predictive models at finer temporal resolutions (Lines 385–393). We also added a supplementary figure with the non-smoothed predictions for sampled localities, as well as a supplementary file (GIF) with the non-smoothed predictions for the entire area. These results were explained briefly in lines 288–292.

I am not convinced of the value of the ‘Niche signal test’ analysis – we know a priori that there are temperature and humidity limits on tick activity and these occur at specific times of the year that correlate with daylength.

Response: We agree with the reviewer that there are conditions of temperature and humidity known to favor tick activity. Our niche signal exploration results, in fact, corresponded to and corroborated such knowledge. We would like to highlight, however, that despite the general correlation between day length and temperature and humidity, the covariance of these variables is not perfectly linear, so exploring their effects is not unjustified. For instance, in the study area, periods with the highest daylengths can have both low or high humidity values. For this reason, we explored all variables that were available to us; as all raw variables allowed us to detect signals of niche, we used them all in our models after transforming them with a PCA.

More minor points:

The article would be improved by a more careful explanation of the objective – i.e. identifying when and where tick activity is occurring. But who would use this information and what for?

Response: We have defined our goal more clearly in the last paragraph of the Introduction (Lines 75–77). We also modified our text to explain the potential uses of the information generated. Lines 81–83. 

Also some alteration of the introduction is needed to explain the focus on A. americanum. PlosOne is an international journal and there are many tick species of public health importance beyond I. scapularis.

Response: We consider that the broad relevance of our contribution is the novel implementation of methods to explore tick activity. We added some words to our introduction to highlight this aspect. Lines 80–81.

The importance of different tick instars as vector of different pathogens should be identified – nymphs and adults (but particularly nymphs which are more numerous than adults) for pathogens with no transovarial transmission, but larvae also for pathogens with transovarial transmission.

Response: We have added text to make this argument completely clear in our Methods section, in the part in which we talk about splitting data according to life stages to create distinct sets of models. Lines 109–110.

Are the numbers of ticks collected at a particular site visit independent of those at the previous visit? If not this needs justification, and if so it needs some kind of accounting in the GLMs.

Response: If we consider tick biology and the periodicity at which two of the ten sites were visited (biweekly), the number of ticks collected in a particular visit may not be independent of those collected during the immediate previous weeks. For this reason, we did not create our models with counts, but rather with information translated into detection (at least one active tick was found) and non-detection (no active ticks were found using the same sampling effort).

I think the comparison of observed tick questing activity and model predictions should be in the main paper.

Response: We agree with the comment. A comparison of model predictions against testing observed data was obtained via model calibration using the metrics sensitivity (prediction rates of activity detected), specificity (prediction rates of activity non-detected), and TSS (a general metric to assess the prediction rates of detections and non-detections). All values for these metrics are now shown in Table 2.

I’m not sure of the value of the mapped ‘extreme conditions’ in Fig 5 as these are only present at some times of the year and don’t describe conditions that result in absence of the ticks.

Response: We have added further descriptions of these results in our text to help readers interpret the importance of visualizing these regions of uncertainty, despite how predictions may look in these areas. Lines 299–301, 304–305.

Figure 1 – the x-axis of time for the bar chart is going right to left rather than the expected left to right.

Response: In the PDF formatted by the journal system, the figure is rotated 90° counterclockwise, which is why the date order appears to be in non-chronological order. Once the figure is rotated, dates are in chronological order from bottom to top.

---

## [Decision Letter · Decision Letter 1]

14 Oct 2024

Modeling spatiotemporal dynamics of Amblyomma americanum questing activity in the Central Great Plains

PONE-D-24-14393R1

Dear Dr. Cobos,

We’re pleased to inform you that your manuscript has been judged scientifically suitable for publication and will be formally accepted for publication once it meets all outstanding technical requirements.

Kind regards,

Ulrike Gertrud Munderloh, Ph.D.

Academic Editor

PLOS ONE

Additional Editor Comments (optional):

Reviewers' comments:

Reviewer's Responses to Questions

**Comments to the Author**

1. If the authors have adequately addressed your comments raised in a previous round of review and you feel that this manuscript is now acceptable for publication, you may indicate that here to bypass the “Comments to the Author” section, enter your conflict of interest statement in the “Confidential to Editor” section, and submit your "Accept" recommendation.

Reviewer #1: (No Response)

2. Is the manuscript technically sound, and do the data support the conclusions?

Reviewer #1: (No Response)

3. Has the statistical analysis been performed appropriately and rigorously? 

Reviewer #1: (No Response)

4. Have the authors made all data underlying the findings in their manuscript fully available?

Reviewer #1: (No Response)

5. Is the manuscript presented in an intelligible fashion and written in standard English?

Reviewer #1: (No Response)

6. Review Comments to the Author

Reviewer #1: (No Response)

7. PLOS authors have the option to publish the peer review history of their article (what does this mean?). If published, this will include your full peer review and any attached files.

Reviewer #1: No

---

## [Editor Report · Acceptance letter]

17 Oct 2024

PONE-D-24-14393R1 

PLOS ONE

Dear Dr. Cobos, 

I'm pleased to inform you that your manuscript has been deemed suitable for publication in PLOS ONE. Congratulations! Your manuscript is now being handed over to our production team.

Kind regards, 

on behalf of

Dr. Ulrike Gertrud Munderloh 

Academic Editor

PLOS ONE